# Operator-theoretic Implicit Neural Representation

## Abstract

The idea of representing a signal as the weights of a neural network, called *Implicit Neural Representations* (INRs), has led to exciting implications for compression, view synthesis and 3D volumetric data understanding. An emergent problem setting here pertains to the use of INRs for downstream processing tasks. Despite a few conceptual results, this remains challenging because the INR for a given image/signal often exists in isolation. What does the local region in the neighborhood around a given INR even correspond to? Based on this inspiration, we offer an operator theoretic reformulation of the INR model, which we call Operator INR (or O-INR). At a high level, instead of mapping positional encodings to a signal, O-INR maps a function space to another function space. A practical form of this general casting of the problem is obtained by appealing to Integral Transforms. The resultant model can mostly do away with Multi-layer Perceptrons (MLPs) that dominate nearly all existing INR models – we show that convolutions are sufficient and offer numerous benefits in training including numerically stable behavior. We show that O-INR can easily handle most problem settings in the literature, where it meets or exceeds the performance profile of baselines. These benefits come with minimal, if any, compromise.

## 1 Introduction

If we view a given signal as a mapping between the domain of measurement to the range space, we can ask if DNNs can help estimate this mapping. One instantiation of this idea is popularly known as *Implicit Neural Representations* (INRs) (Sitzmann et al., 2020; Tancik et al., 2020; Mildenhall et al., 2021; Fathony et al., 2021) which can parameterize spatial/spatio-temporal data (Gropp et al., 2020; Niemeyer et al., 2019; Jiang et al., 2020) for applications in image super-resolution (Chen et al., 2021), texture synthesis (Oechsle et al., 2019), inverse problems (Sun et al., 2021; Yu et al., 2021b; Niemeyer et al., 2020), and novel view synthesis (Mildenhall et al., 2021; Sun et al., 2022).

**From one signal to a set of signals.** INRs typically consist of a neural network that is trained to map each coordinate of a given signal's domain to its measurements/values, and so also known as *coordinate-value networks*. The mapping is learned via a neural network and gives a compact representation of the signal (Sitzmann et al., 2020; Fathony et al., 2021). The discussion above was in the context of *one signal (or image)*. When given a *set* of signals, one approach is to derive an INR for each signal in the set – Dupont et al. (2022a) then uses these functions (called functa) as data for downstream deep learning tasks. Alternatively, one can estimate a meta-learned "base" (INR) network, and associate each data sample (or signal) in the dataset as a *modulation* of the base network (Dupont et al., 2022b), akin to random effects modulating fixed effects in mixed effects models (Lindstrom & Bates, 1990). The modulation can also be accomplished in other ways as we will see later (Feng et al., 2022), via introducing a surrogate vector which is tied to a specific INR through conditioning. Now, for a simplified case, where the data samples were *ordered* with respect to a surrogate variable, we get a set of INRs where

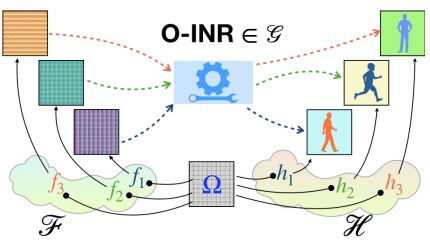

Figure 1: Overview of O-INR: $f_1, f_2, f_3 \in \mathcal{F}$ are input functions acting on the domain $\Omega$. O-INR maps these functions to their corresponding signals (functions) $h_1, h_2, h_3 \in \mathcal{H}$.

each sample (or signal) specific INR is a level set with discrete values denoting the levels. Here, we take this interpretation to the extreme to check its advantages, see Fig. 1.

**This paper.** A prevailing view is to consider the INR as a coordinate-value transform. We study a generalization – one where we still wish to parameterize a signal (i.e., an identical goal as INR) but as a transformation between two *function spaces*. Casting INRs in this manner yields an *operator-theoretic* view: our object of interest is the operator that takes us across function spaces. We model these transforms via *integral operators* (or integral transforms) which, by definition, transform between function spaces via the process of integration. If we further constrain the integral operator to be local and translation-equivariant, we arrive at an efficient parameterization in terms of convolutional layers. Apart from its succinctness and simplicity, our goal is to show how this formulation gives rise to many benefits compared to coordinate-based INRs.

**Contributions.** **(a)** We introduce a new type of INR called Operator INR (O-INR) which yields comparable or superior empirical performance relative to common methods in terms of representation capability on 2D images and 3D scenes; **(b)** While most INR parameterizations rely on large MLPs, we show that convolution operations with sinusoidal non-linearities are more efficient to train and faster to evaluate. **(c)** Higher order derivatives of O-INRs can be efficiently computed in closed form, allowing efficient processing in downstream tasks such as denoising. **(d)** O-INR offers greater convenience (control over both the input function space and the weight space), including explicit control of the spatial interpolation behavior, mitigating the influence of initialization, and more interpretable behavior in weight space.

## 2  RELATED WORK

**Implicit Neural Representations:** INRs are useful in a wide range of tasks. By virtue of learning a continuous mapping, INRs can be sampled at any resolution thereby making them applicable in super-resolution and denoising (Saragadam et al., 2022; 2023; Peng et al., 2020). Other tasks including 3D rendering, boundary value problems, PDEs and generative modeling (Skorokhodov et al., 2021; Esmaeilzadeh et al., 2020; Schwarz et al., 2020) have also been studied using INRs. In (Shaham et al., 2021), the authors leveraged INRs for high resolution image to image translation. While the original development of INRs was intended for Euclidean data, more general non-Euclidean domains have been studied recently as in (Grattarola & Vandergheynst). Many works have also adapted INRs for scene representation (Niemeyer & Geiger, 2021; Guo et al., 2020; Yu et al.) and scene editing (Yuan et al., 2022; Feng et al., 2022; Fan et al., 2022; Gong et al., 2023).

Recall that the original formulation of an INR is as a multilayer perceptron (MLP). Various reparameterizations have been developed that seek to offload computation to other components to enable faster training and inference, especially in the context of their use in NeRFs. For example, plenoxels (Fridovich-Keil et al., 2022) and plenoctrees (Yu et al., 2021a) represent a radiance field with explicit voxel or octree structures and DirectVoxGO (Sun et al., 2022) stores features on a voxel grid which is decoded into radiance values with a tiny MLP.

Several approaches have been proposed for learning in the context of downstream tasks using INRs (Wang & Golland, 2022; Xu et al., 2022; Dupont et al., 2022a;b). The use of differential operators on INRs has been demonstrated in (Xu et al., 2022) by treating INRs as functions which enables modification without explicit decoding. Further, in Wang & Golland (2022), the authors treat neural fields as integrable maps and propose discretization invariant layers that map elements of this function space to be readily used in DNN models.

**Continuous convolutions:** Since discrete convolutions learn weights which are tied to the relative positions, continuous convolutions were initially designed to handle irregularly sampled data (Schütt et al., 2017; Simonovsky & Komodakis, 2017; Wu et al., 2019). Continuous time convolutions are well studied, but their recent use in deep learning applications includes modeling point clouds (Wang et al., 2021; Boulch, 2019), graphs (Fey et al., 2017), fluids (Ummenhofer et al., 2019), and even sequential data (Romero et al., 2021b;a). We note that the use of convolutions within INRs is rare (Peng et al., 2020). In most settings above, irregular sampling intervals can be handled while maintaining locality and translation invariance. CNNs for modeling long range dependencies in arbitrary number of dimensions have also been studied (Romero et al., 2022).

## 3   SETTING UP O-INRS

We denote the standard coordinate-valued network as $m_\theta : \mathbb{R}^D \to \mathbb{R}^R$ where $\theta \in \mathbb{R}^N$ denotes the parameters of the neural network, usually based on multi-layer perceptrons (MLPs). Here, $D$ (and $R$ resp.) denote the dimensionality of the domain (and the range space resp.) of the continuous function being learned. For example, when fitting an INR to a 2D RGB image, $D = 2$ and $R = 3$.

**The Space of Discretized Positional Encodings:** It is well documented that coordinate-value networks fail when coordinate locations are directly provided as input (Tancik et al., 2020; Sitzmann et al., 2020), and various strategies have been developed to lift the coordinates to a higher-dimensional representation more conducive to MLP fitting. These *positional encodings* on the coordinate space typically involve sinusoids across a range of frequencies (Mildenhall et al., 2021). For example, consider the encoding:

$$f(\llbracket x, y \rrbracket) = [\sin(x), \cos(x), \sin(y), \cos(y), \ldots, \sin(2^L x), \cos(2^L x), \sin(2^L y), \cos(2^L y)] \quad (1)$$

We have many choices for this input, of the form $f(\llbracket x, y \rrbracket) = [\sin(\theta x), \cos(\theta x), \sin(\theta y), \cos(\theta y)]$, where $\theta$ can even be a tunable parameter as in Zhou et al. (2021). In fact, it even makes sense to consider the entire family of such functions, say by varying $\theta$, which share a common domain and co-domain. The corresponding space we will obtain is commonly referred to as a *function space*.

More importantly, for a sequence of signals defined on the *same* domain, $\Omega$, the corresponding positional encodings $f_1, \cdots, f_n$ also act on $\Omega$. These are different functions but belong to the same family w.r.t. their domain of definition, regularity properties, and so on. We must interpret this set to correspond to a well defined space of functions.

Even in the simplest case of two functions $f_1, f_2 \in \mathcal{F}$, the mapping via standard INRs will be independent. We will obtain two *separate* models. However, both are members within the family of functions discussed above. And due to how these functions are defined, there is extensive structure in the function space that can be utilized. Based on these observations, we define our input function space in terms of sinusoidal positional encodings (Sitzmann et al., 2020). Specifically,

$$\mathcal{F} = \{f | f : \Omega \to \psi\} \quad (2)$$

where $\Omega$ is the domain of definition, e.g., 2D plane for images, and a 3D cube for volumes whereas $\psi$ defines the space of sinusoidal positional encodings.

**Signal spaces:** In the above discussion, the elements of the function space $\mathcal{F}$ were considered to belong to the family of sinusoidal embedding functions. But INRs learn a map from the positional encoding space to the signal space. So, what do $f_1, f_2, \cdots, f_n$ yield after going through such a map? The answer is clear when we think of the associated co-domain of this map simply as another function space. We denote this function space of signals by $\mathcal{H}$, also defined on the domain $\Omega$. Elements of $\mathcal{H}$ namely, $h_1, h_2, \ldots, h_n$ are essentially the different signals (for example, frames in a video) whose corresponding embedding in $\mathcal{F}$ is $f_1, f_2, \ldots, f_n$.

**From INRs to O-INRs:** Many tasks can be posed as learning a map between two function spaces $\mathcal{F}$ and $\mathcal{H}$. We parameterize the transformation between these function spaces via a neural network (Rosasco et al., 2010; Que et al., 2014):

$$\mathcal{G}_\phi : f \to h \quad (3)$$

where $\phi$ represents the parameters of a DNN, and $f \in \mathcal{F}$ and $h \in \mathcal{H}$ are functions. We refer to this operator based formulation of an implicit representation as O-INR. While (3) gives a very general transform, we need a little more structure on the operator to allow efficient learning.

**Integral operators:** Let us assume a simplified setup. We want to learn a map from $f_1 \to h_1$, where $f_1 \in \mathcal{F}$ and $h_1 \in \mathcal{H}$. Consider the common domain to be $\Omega = \mathbb{R}$, the 1D real number line. The simplest map would be an identity mapping, resulting in $h_1(x) = f_1(x), \forall x \in \Omega$. At the other extreme, we can write $h_1(x) = \mathcal{C}(x, \{f_1(y) | y \in \Omega\})$, where the value of $h_1(x)$ depends on all evaluations of $f_1$ via a functional $\mathcal{C}$. This can be written as an integral *along* the domain $\Omega$,

$$h_1(x) = \int_{y \in \Omega} \mathcal{C}(x, f_1(y)) \mathrm{d}y \quad (4)$$

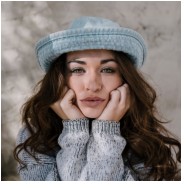 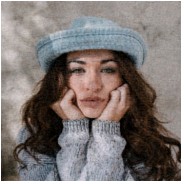 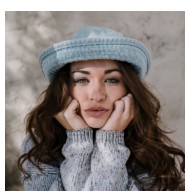 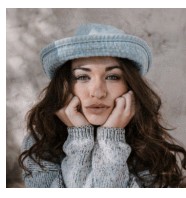 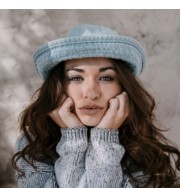

Ground truth    **O-INR** (26.05)    SIREN (28.56)    WIRE (26.44)    MFN (29.36)

Figure 2: Performance comparisons of O-INR in multi-resolution training setting. The ground-truth together with images from O-INR and other baselines (L to R), with the PSNR value in dB. O-INR achieves comparable performance

This means that an integral operator achieves the transformation between the function spaces via integrating over the domain of definition. This is beneficial: since integral operators are defined using their associated kernels, the only parameterization we need within O-INR will be this kernel!

Consider $f \in \mathcal{F}$ and $h \in \mathcal{H}$ as functions over the domain $\Omega$, we learn an integral operator $\mathcal{G}_\phi$ with the associated kernel $\mathcal{K}_\phi$, where $\phi$ denotes the parameterization involved. Then, the integral transform can be represented as:

$$h(\omega) = \mathcal{G}[f](\omega) = \int_{\omega' \in \Omega} \mathcal{K}_\phi(\omega, \omega') f(\omega') \mathrm{d}\omega', \quad \omega \in \Omega \tag{5}$$

where $\mathcal{G}[f]$ denotes the application of the transform on the function $f$. Note that we recover the behavior of a standard coordinate-valued network if the kernel is modulated by a Dirac delta function: $\mathcal{K}_\phi(\omega, \omega') = K_\phi(\omega)\delta_{\omega'}(\omega)$. In which case, we have $h(\omega) = K_\phi(\omega)f(\omega) = \tilde{K}_\phi(f(\omega))$.

**How to parameterize O-INR?** From (5), the *only* parameterization in our formulation is through $\mathcal{K}_\phi$. In its maximum capacity, the bi-variate function $\mathcal{K}_\phi$ can take distinct parameters for each pair of distinct $(\omega, \omega')$. While nearly all INR formulations perform pointwise evaluations with an MLP decoder, we can take advantage of our model and use convolution layers to parameterize O-INR. Considering the associated kernel to be a convolutional kernel, we have: $\mathcal{K}_\phi(\omega, \omega') = g_\phi(\omega - \omega')$. Therefore, with $g_\phi$ being the standard convolutional kernel (5) becomes:

$$h(\omega) = \mathcal{G}[f](\omega) = \int_{\omega' \in \Omega} g_\phi(\omega - \omega') f(\omega') \mathrm{d}\omega', \quad \omega \in \Omega \tag{6}$$

Notice that in standard INRs, the mapping is a point-wise map, hence in the latent space (of INRs), adjacency does not have a semantic correspondence with the spatial dimension. But in O-INRs, the transform is obtained over the entire domain of definition and hence the use of location bias is permissible.

**Multi-resolution training & Continuous convolutions:** *How to sample at arbitrary resolution?* When using convolution kernels to parameterize O-INR, one drawback emerges when we want to sample the signal at *any arbitrary* resolution. This is because discrete convolutions cannot adapt their weights to different spacings, resulting in poor performance when changing resolution. A remedy is available via continuous convolutional kernels (Romero et al., 2021b). We also parameterize each INR as a continuous convolutional network, which is trained to map multiple resolutions of positional encodings of the domain of definition of the signal (e.g., the 2D plane) to corresponding resolutions of the desired signal (e.g., images).

**Remark 1** *For spatio-temporal data the dimension of time is treated differently in the context of the positional encoding as we will discuss in §5.*

**Remark 2** *While the rationale of positional encoding is to provide high frequency signals as inputs to the model, our use of convolutional layers also makes it possible to simply use noise as a proxy for the high frequency positional encoding term. But this is a poor choice within INRs with MLP layers due to the lack of location bias.*

**Miscellaneous implementation details:** While we use the aforementioned positional encoding in our experiments, we also train O-INR with noise as an additional channel to provide high frequency

components and it works well. For ease of implementation, and in cases where the sole purpose is to fit to one resolution of the data point, we can use discrete convolutions. We now discuss of our experimental results using continuous convolutions next.

## 4 REPRESENTATION CAPABILITY OF O-INR

We first check the representation capability of O-INRs relative to standard INRs. We evaluate performance on 2D images as well as 3D volumes. Additionally we show that our proposed model can handle inverse problems such as image denoising. For 2D images, we use images from several sources including Agustsson & Timofte (2017) Kodak Image Suite, scikit-image, etc. For 3D volumes, we use data from the Stanford 3D Scanning Repository and Saragadam et al. (2022; 2023).

### 4.1 MULTI-RESOLUTION TRAINING IS POSSIBLE

***Task.*** We will assess the effectiveness of the multi-resolution training approach for O-INR. Given an image at a particular resolution, we train our model using its lower resolution versions (obtained by down sampling). Can our model effectively reconstruct images at an arbitrary resolution?

***Setup.*** We compare our method to baselines including SIREN (Sitzmann et al., 2020), WIRE (Saragadam et al., 2023) and MFN (Fathony et al., 2021). Following (Saragadam et al., 2023), we train the baselines on the best resolution image seen by the O-INR during training. We then compare performance of all methods for reconstructions at the original (higher) resolution. Note that O-INRs with continuous convolutions can be trained at multiple resolutions.

***Results summary.*** As seen in Fig. 2, O-INR achieves comparable/better performance than baselines in terms of the Peak Signal to Noise Ratio (PSNR). Due to the use of continuous convolutions here, the number of parameters required for O-INR are much smaller (100K) compared to baseline models ($\sim 130K$) to achieve parity in performance.

### 4.2 2D IMAGE REPRESENTATION EFFECTIVENESS

***Task.*** A prominent use case of INRs is in representing spatio-temporal signals. So, is O-INR effective at representing 2D images of varying resolutions?

***Setup.*** We compare O-INR with SIREN (Sitzmann et al., 2020), WIRE (Saragadam et al., 2023) and MFN (Fathony et al., 2021) based on **(i)** PSNR and **(ii)** training time to reach the best possible PSNR for that specific model. While other INR models use MLP layers, our model is solely parameterized by convolution layers. For fair comparisons, the number of parameters in our models are comparable with the baselines.

***Results summary.*** Table 1/Fig. 3 shows that O-INR achieves comparable/better performance than baseline methods in terms of PSNR. In terms of training time, O-INR is comparable/better than MFN/WIRE but slower than SIREN (but in all cases, SIREN representations seem to underperforms the other baselines).

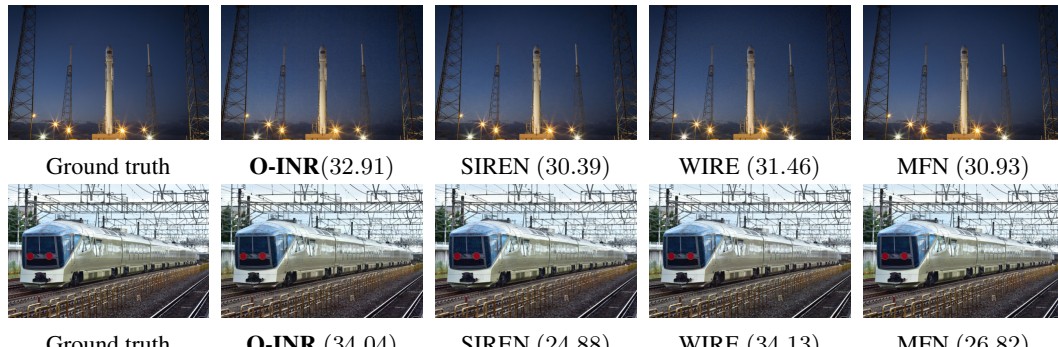

| Ground truth | **O-INR**(32.91) | SIREN (30.39) | WIRE (31.46) | MFN (30.93) |
| Ground truth | **O-INR** (34.04) | SIREN (24.88) | WIRE (34.13) | MFN (26.82) |

Figure 3: Performance comparison of O-INR for 2D image representation. Each row displays the ground-truth together with images from O-INR and other baselines (L to R), with the PSNR value in dB. O-INR achieves comparable/better performance.

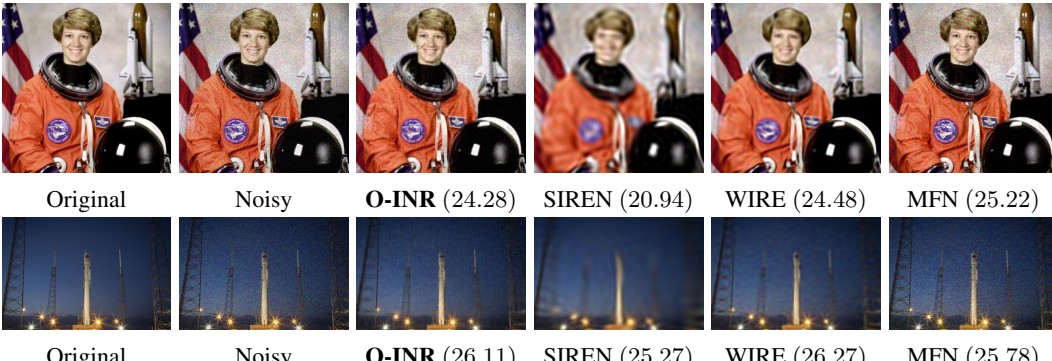

| Original | Noisy | **O-INR** (24.28) | SIREN (20.94) | WIRE (24.48) | MFN (25.22) |

| Original | Noisy | **O-INR** (26.11) | SIREN (25.27) | WIRE (26.27) | MFN (25.78) |

Figure 4: Performance comparisons of O-INR for representing noisy images. For each method, we show the PSNR for the image in dB. Among all methods, SIREN achieves the lowest PSNR, O-INR and other baselines perform similarly.

## 4.3 APPLICATION TO IMAGE DENOISING

***Task.*** Our task is to assess the robustness of O-INR: is it effective at representing noisy images?
***Setup.*** Given an image, following (Saragadam et al., 2023), we add photon noise for each pixel via independently distributed Poisson r.v. (maximum mean photon count 30, readout count 2). These noisy images are then used to learn O-INR models. We compare performance with SIREN, WIRE and MFN. Consistent with previous experiments, all models have comparable number of parameters.

***Results summary.*** From Tab.2/Fig. 4, we see that O-INR is able to nicely recover the true signal comparable with other methods. This experiment (also see appendix) indicate its effectiveness in solving some inverse problems.

## 4.4 3D VOLUME REPRESENTATION

***Task.*** INRs are commonly used as a continuous representation of 3D volumes or surfaces. So, can O-INR encode 3D volumetric data well?
***Setup.*** We consider occupancy volume sampled over a $512 \times 512 \times 512$ voxel grid, where each voxel *within* the volume is assigned a value of 1 inside an object and 0 otherwise. We compare O-INR with SIREN, WIRE and MFN based on intersection over union (IoU). We ensure a similar number of parameters when comparing with baselines.
***Results summary.*** Fig. 6 shows that O-INR performs well in IoU in all cases. For SIREN and WIRE, we report the best performance (achieved with a model with slightly fewer number of parameters). Increasing the parameters of SIREN and WIRE involves an interplay with other hyper-parameters.

|      |       | Train | Fern | Coffee | Walnut | Rocket |
|------|-------|-------|------|--------|--------|--------|
|      | Size  | $510 \times 339$ | $510 \times 339$ | $400 \times 600$ | $510 \times 339$ | $427 \times 640$ |
| PSNR | SIREN | 24.88 | 28.27 | 29.91 | 26.09 | 30.39 |
|      | WIRE  | 34.13 | 37.16 | 31.51 | 33.21 | 31.46 |
|      | MFN   | 26.82 | 31.26 | 32.27 | 28.36 | 30.93 |
|      | O-INR | 34.04 | 36.4 | 32.04 | 32.12 | 32.91 |
| Time | SIREN | 38.43 | 53.8 | 68.67 | 51.3 | 89.39 |
|      | WIRE  | 100.36 | 109.41 | 151.27 | 102.24 | 198.49 |
|      | MFN   | 89.7 | 88.05 | 117.76 | 82.06 | 151.99 |
|      | O-INR | 88.0 | 68.2 | 101.3 | 64.83 | 109.63 |

Table 1: O-INR and baselines for 2D image representation. PSNR (in dB) and time in seconds show O-INR is comparable/better than baselines.

## 5 REPRESENTING A SEQUENCE OF SIGNALS/FUNCTIONS

***Task.*** Given a sequence of signals captured over a predefined fixed domain, it is natural to consider the data as a sequence of functions defined over the domain yielding a sequence of functions (or signals). For example, frames in a short-burst video are a sequence of images (captured by different functions over the same domain). In standard INR formulations, such signals are represented by considering an additional parameter (usually time) in the domain of definition and parameterized using $m_\theta : (x, y, t) \to (r, g, b)$. While this is reasonable, a more natural approach from a operator (functional) perspective is to consider the sequence of frames as different (but related) functions acting on the same domain rather than a function (with a rather large

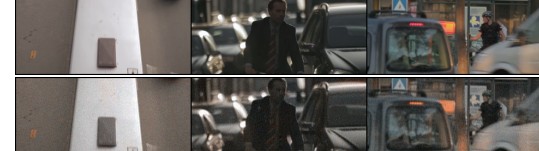

Figure 5: Top/Bottom: Rows show frames from a bike video: original and the ones from O-INR trained on sparsely sampled frames of a long video sequence. O-INR represents the scenes in the sequence well.

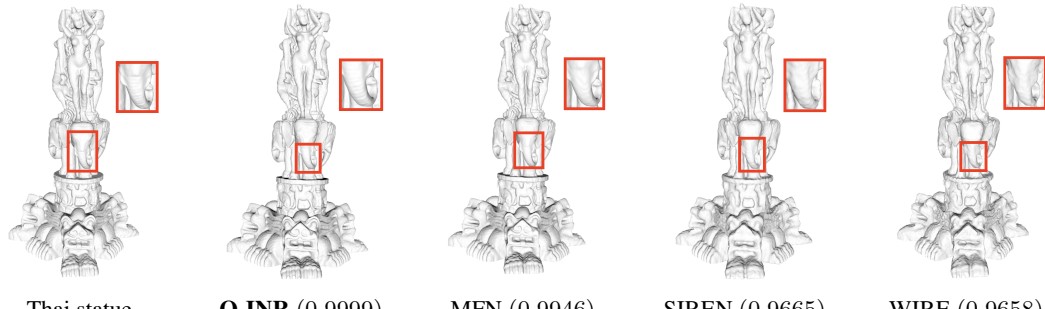

Thai statue     **O-INR** (0.9999)     MFN (0.9946)     SIREN (0.9665)     WIRE (0.9658)

Figure 6: We report IoU achieved for each method after training converged. O-INR achieves best performance among all baselines. Zoomed in parts in each case show the that minute details are captured better by O-INR.

redundancy) acting on the spatio-temporal volume. Our experiment checks if O-INR is effective here.

**Setup.** We consider learning a transform between spaces consisting of sequence of functions. More precisely, in this case O-INR takes the following form:

$$\mathcal{G}_\phi : \mathcal{F}_N \to \mathcal{H}_N; \quad \mathcal{F}_N = \{f_n(\Omega)|n \in \mathbb{N}\} \quad \mathcal{H}_N = \{h_n(\Omega)|n \in \mathbb{N}\} \tag{7}$$

where $\phi$ denotes the to-be-learned parameters and $\Omega$ is the domain of definition. We use sinusoidal positional encodings as the input function space. A key question here is how to define a sequence of functions over the domain under consideration, while still ensuring that all such functions provide both low and high frequency signals as an input to our O-INR. Here, we consider the domain $\Omega$ as the 2D plane over which frames are defined, with $(x, y) \in \Omega$ and $\gamma = \alpha + ((\beta - \alpha)/N)n$

$$f_n([\![x, y]\!]) = \left[\sin(2^l\pi x) + \gamma, \cos(2^l\pi x) + \gamma, \sin(2^l\pi y) + \gamma, \cos(2^l\pi y) + \gamma, \ldots, \right] \tag{8}$$

where $l \in [0, L-1]$ for some $L \in \mathbb{N}$, levels of frequencies chosen to be part of the input signal. Here, $\alpha, \beta$ are empirically determined constants and $N$ is the total number of functions that the O-INR is trained to encode.

**Results summary.** In the first experiment, we learn O-INR for consecutive frames of a video of a cat from SIREN (Sitzmann et al., 2020). In the second case, we train O-INR effectively on a much longer sequence from skvideo-dataset, albeit only using sparsely sampled frames, thereby exploiting the regularity property of the function space associated with a related sequence of signals. From Fig. 5 and Fig. 7, we see that O-INR can faithfully represent the different frames via a single model. We find that these results are comparable with SIREN (appendix includes more examples).

## 6   APPLICATIONS TO BRAIN IMAGING FOR SLICE IMPUTATION

**Task.** Available software tools (like FreeSurfer (Fischl, 2012)) for the analysis of brain imaging data target high resolution scans, acquired within research studies. However, as noted in (Dalca et al., 2018), typical clinical (non-research) scans have much lower out of plane resolution (often for slice-by-slice reading by radiologists). Processing such scans with existing tools poses difficulties, and the results often need to be manually checked. Even partially mitigating this issue can radically increase sample sizes available for scientific analysis. We demonstrate the use of O-INR in representing such low resolution brain imaging data in (a) obtaining a faithful representation and (b) preserving statistical group differences.

| | Astronaut | Cat | Kodak05 | Kodak19 | Rocket |
|---|---|---|---|---|---|
| Size | $512 \times 512$ | $300 \times 451$ | $512 \times 768$ | $256 \times 171$ | $427 \times 640$ |
| SIREN | 20.94 | 24.8 | 18.27 | 22.13 | 25.27 |
| WIRE | 24.48 | 27.4 | 20.47 | 24.6 | 26.27 |
| MFN | 25.22 | 24.8 | 23.6 | 20.96 | 25.78 |
| O-INR | 24.28 | 25.1 | 21.9 | 22.7 | 26.11 |

Table 2: Comparison of PSNR values (in dB) for O-INR and baselines for 2D image denoising.

**Setup.** We consider MRI data from Alzheimer's Disease Neuroimaging Initiative (ADNI) (Mueller et al., 2005; Jack Jr et al., 2008) and model 2D slices in a 3D brain scan via a sequence of functions as described in §5. Our data includes approximately 140 subjects each from cognitively normal (CN) and diseased (AD) individuals.

O-INRs are trained for each image. For each 3D image, we progressively dropped more slices in one out of plane direction to simulate poor resolution, and use it for training.

***Results summary.*** We observe that O-INR remains robust up to a high percentage of missing slices. The MSE of the reconstructed brain volumes reported in Tab. 3 show that O-INR is capable of learning the representation well enough even when more than $80\%$ of the slices are dropped. Finally, we use SnPM (Statistical NonParametric Mapping) toolbox (Ashburner, 2010) to perform a statistical group difference analysis (voxel-wise $t$-test) on the real data (CN versus AD). Then, the same analysis was performed on O-INR derived data (CN versus AD). We find that voxels reported to be significant (uncorrected $p$-values) on the real data analysis

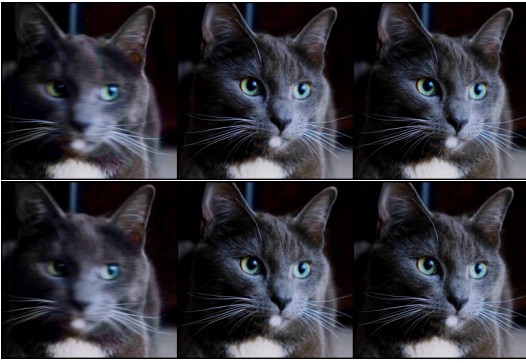

Figure 7: Top to Bottom: Rows represent frames from cat video (Sitzmann et al., 2020) original and ones obtained from O-INR representation. In this case the model was trained on consecutive frames of the video.

overlap with the analysis on data based on O-INR slice imputation. Sizable clusters agree although the spatial extent is reduced (higher Type 2 error). Statistical analysis results with a brain image underlay and additional results from the group difference analysis are given in Appendix §I.

## 7   LEARNING DOWNSTREAM TASKS ON O-INRS

***Task.*** Using INRs for downstream tasks is an exciting emergent problem setting. Recently (Navon et al., 2023) proposed an equivariant architecture for learning in the so-called "deep weight spaces" of INRs. However, in general, with standard INRs, signal processing operations on the latent space of MLPs remain difficult (Xu et al., 2022). Most methods must resort to discretization leading to loss in properties like continuity. The result in (Xu et al., 2022) explores the use of differential operators on INR: it is interesting but is not memory efficient (see pp 10 (Xu et al., 2022)). Since the O-INR operates on function spaces, many operations in signal processing (e.g., evaluating derivatives) are incredibly easy in principle. So, can these benefits be verified in practice?

| % missing | Train MSE | Test MSE |
|-----------|-----------|----------|
| 50 | 2.17e-5 | 8.33e-5 |
| 66 | 3.10e-5 | 2.38e-4 |
| 75 | 1.39e-6 | 4.70e-4 |
| 80 | 1.58e-5 | 7.66e-4 |
| 82 | 2.69e-6 | 1.08e-3 |

Table 3: Train/test MSE for 3D brain images with % of missing slices during training O-INR.

***Setup.*** When using O-INR to encode a signal e.g., an image, the signal is represented as the convolution of a known simple signal (e.g., a deterministic positional encoding) with a sequence of learned kernels. For ease of presentation, consider the domain to be one dimensional. Then our model is:

$$h(x) = f(x) * g(x) \tag{9}$$

where $h(x)$ is the true signal we want to represent, $f(x)$ is the positional encoding and $g(x)$ is the learned transform (convolution filters) between $f(x)$ and $h(x)$. When taking into account our multi-layer convolutional model with sine non-linearities, (9) can be written as (e.g., for 3 layers):

$$h(x) = (\sin(\sin(f(x) * g_1(x))) * g_2(x)) * g_3(x) \tag{10}$$

We make use of the property of computing derivatives over the convolution operation, namely:

$$h(x) = f(x) * g(x) \implies h'(x) = f'(x) * g(x) \tag{11}$$

Then, the derivative of our original signal is:

$$h'(x) = (\sin(\sin(f(x) * g_1(x))) * g_2(x))' * g_3(x) \tag{12}$$

which on repeated application of equation 11 and the chain rule leads to $\cos(\sin(f(x) * g_1(x))) * g_2(x)) \odot \cos(f(x) * g_1(x)) \odot f'(x) * g_1(x) * g_2(x) * g_3(x)$, where $\odot$ denotes point-wise multiplication and $*$ denotes convolution operation. The extension to higher order derivatives follows similarly.

***Results summary.*** We show the effectiveness of this approach in computing derivatives in Fig. 8. We see that once an O-INR is learned, it can map different functions to their desired signals. Here,

the input functions are the positional encodings of the grid and its derivative, which are then mapped via O-INR to their corresponding outputs: signal (image) and its first-order gradient. This shows that O-INR allows seamless calculus operations in the function space, *a functionality not currently available otherwise*.

# 8  O-INR WEIGHT INTERPOLATION

***Task.*** We verified above that operations like derivatives are possible, suggesting that structure in $\mathcal{G}$ can be queried. So, does this ability allow other operations (interpolation)?

***Setup.*** We investigate whether the convolutional weight space of O-INRs produces a more structured latent space than coordinate-based networks. One way to do this is by visualizing interpolations between O-INRs fit on different images from the CelebA dataset (Liu et al., 2015). We should note that no generative model was trained on CelebA – the *only* two images that O-INR sees are the ones being interpolated.

***Results summary.*** Ainsworth et al. (2022) recently demonstrated the use of special weight-matching algorithms to align two models in the weight space. Using this idea, we permute the channel ordering of one O-INR's layers to minimize the total cosine distance between the activation statistics of the two O-INRs. Interpolation results presented in Fig. 9 use this strategy. Interestingly, even without an explicit weight matching, we find that all trends hold. We find that performing linear interpolation between the convolutional weights (in O-INR) corresponding to different images leads to reasonable and interpretable outputs, whereas interpolating individual layers in coordinate-based MLPs like SIREN does not yield coherent outputs Fig. 9. More examples of such interpolations and results obtained by manipulating individual convolutional layers are provided in Appendix §J.

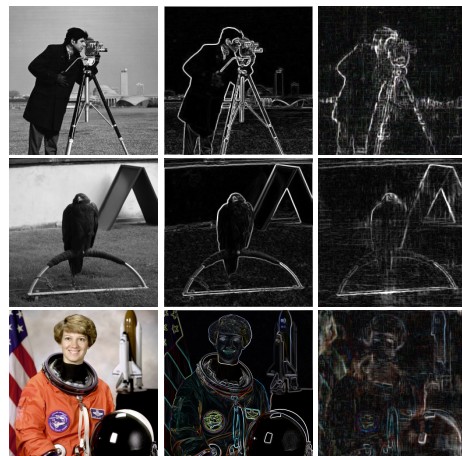

Figure 8: (L to R) Original image, true gradient of the image via Sobel filter and gradient obtained via O-INR (see §7). We see that the O-INR derivative closely matches the true derivative in all cases. Small discrepancies are due to the residual between the true image and its O-INR representation.

# 9  CONCLUSION

O-INR constitutes the first approach for fitting INRs that treats coordinate encodings as a function space, giving rise to efficient and compact training on complex signals and particularly sequences of signals. O-INR also leverages the properties of convolutional neural

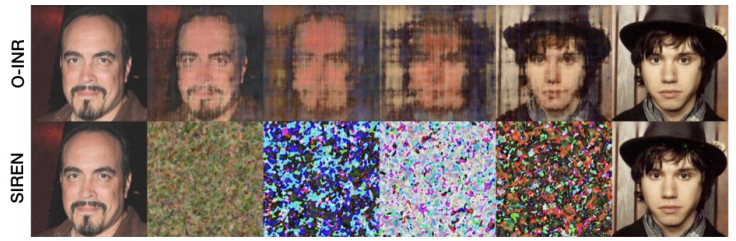

Figure 9: Interpolations between two CelebA images fit with SIREN versus O-INR. All layer weights are *linearly* interpolated.

networks and sinusoidal activations to produce fast closed form derivatives useful in downstream tasks as well as an interpretable latent weight space. We observe that O-INR is effective on a wide range of signals and tasks, and requires little to no hyperparameter tuning. Future work will expand on the possibilities opened by this framework, such as fitting a radiance field with a single interpolation-free CNN whose input captures all the information about camera rays and their query points. We also hope to address limitations of this approach, particularly maintaining high performance and speed on arbitrary non-grid inputs, quite important in many 3D applications.

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

## A    APPENDIX

We present additional experimental details and empirical results for the different experiments presented in the main paper and some ablation studies, pertaining to the use of noise as a channel for positional encoding.

## B    MULTI-RESOLUTION TRAINING FOR 2D IMAGES

For each image we trained O-INR on a bunch of lower resolution images. For example: the cameraman image was originally of size $256 \times 256$, and hence O-INR was trained using images in the range $120 \times 120$ to $224 \times 224$. Similarly, for the image of human face O-INR was trained on images with size in the range $256 \times 256$ to $360 \times 360$ and final performance was evaluated on the original image of size $512 \times 512$. Since, baselines such as SIREN Sitzmann et al. (2020), WIRE Saragadam et al. (2023) and MFN Fathony et al. (2021) can only be trained on image of one resolution, we first trained them on the lowest, highest and average resolution of image used for training O-INR, however as expected the performance of baseline models was the best when they were trained using the highest resolution of image used to train O-INR. Hence, we report PSNR for all baseline methods trained on the highest resolution of image used to train O-INR. The performance is measured on the original (higher) resolution image in all cases.

For training O-INR we used a learning rate of 0.0005 for 1000 epochs and the number of sinusoidal frequencies used for each dimension was 20, 10 coming from sin and 10 from cos. The number

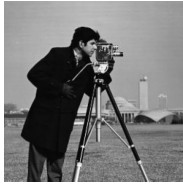 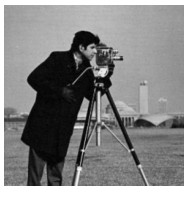 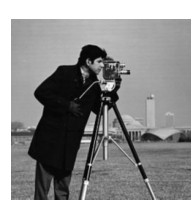 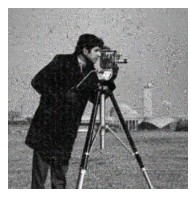 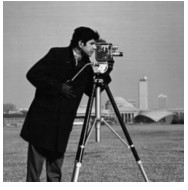

Ground truth   **O-INR** (32.14)   SIREN (31.84)   WIRE (22.1)   MFN (32.67)

Figure 10: Performance comparisons of O-INR in multi-resolution training setting. The ground-truth together with images from O-INR and other baselines (L to R), with the PSNR value in dB. O-INR achieves comparable/better performance

of parameters for O-INR model to achieve comparable performance was $\approx 100$k whereas baseline methods required $\approx 130$k parameters. Additional results are presented in Fig. 10

## C 2D IMAGE REPRESENTATION

For 2D image representation, O-INR and all other baseline methods were trained on the image of same resolution and representation capability was presented in terms of PSNR. All models had $\approx 172$k trainable parameters and trained until convergence with learning rate in the order of $0.001$. Here, the number of sinusoidal frequencies used for each dimension was 20, 10 coming from $\sin$ and 10 from $\cos$. In Fig. 11, we present additional results for 2D image representation.

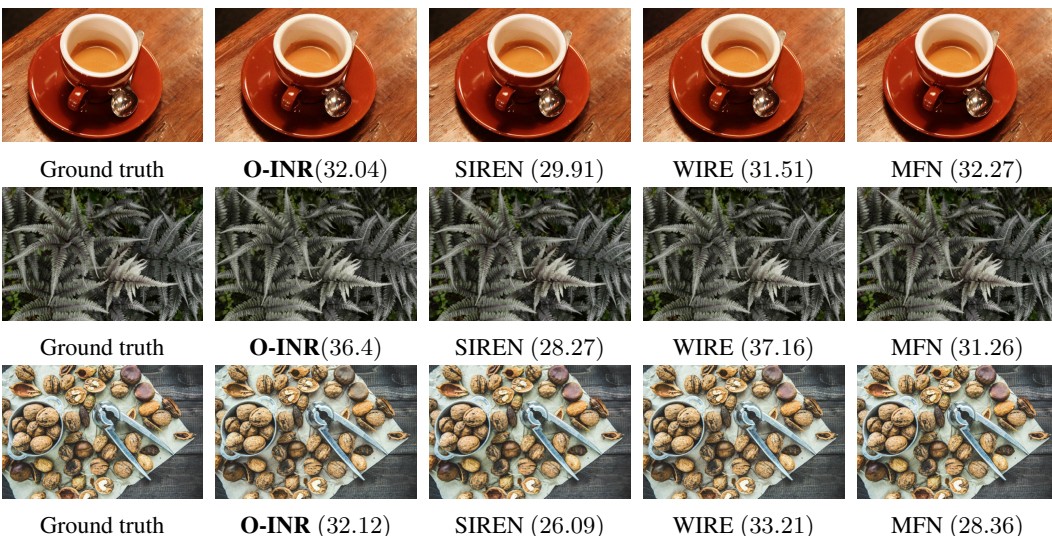

Figure 11: Performance comparison of O-INR for 2D image representation. Each row displays the ground-truth together with images from O-INR and other baselines (L to R), with the PSNR value in dB.

## D 3D VOLUME REPRESENTATION

For 3D volume representation both O-INR and baseline models were trained with $\approx 1$M parameters. We used a learning rate of $0.001$. In this case, the number of sinusoidal frequencies used for each dimension was 16, 8 coming from $\sin$ and 8 from $\cos$. Additional result in Fig. 12 shows that in terms of IoU, O-INR performs as good as the alternative.

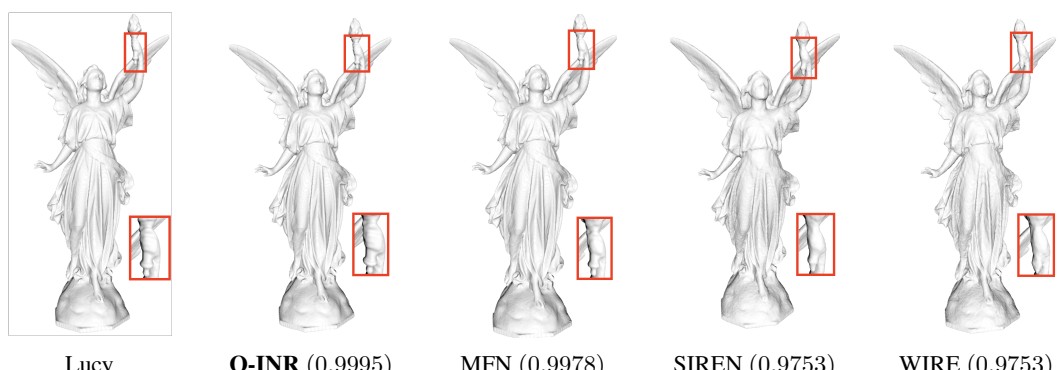

Figure 12: We report IoU achieved for each method after training converged. O-INR achieves best performance among all baselines. Zoomed in parts in each case show the that minute details are captured better by O-INR.

# E  2D IMAGE DENOISING

In recovering an image from a noisy version we trained O-INR and all baseline models on noisy variants obtained by following the noise addition procedure in Saragadam et al. (2023). All models had roughly 130k trainable parameters. We used a learning rate of 0.003 for O-INR. Here, the number of sinusoidal frequencies used for each dimension was 20, 10 coming from sin and 10 from cos. In Fig. 13 we present additional results.

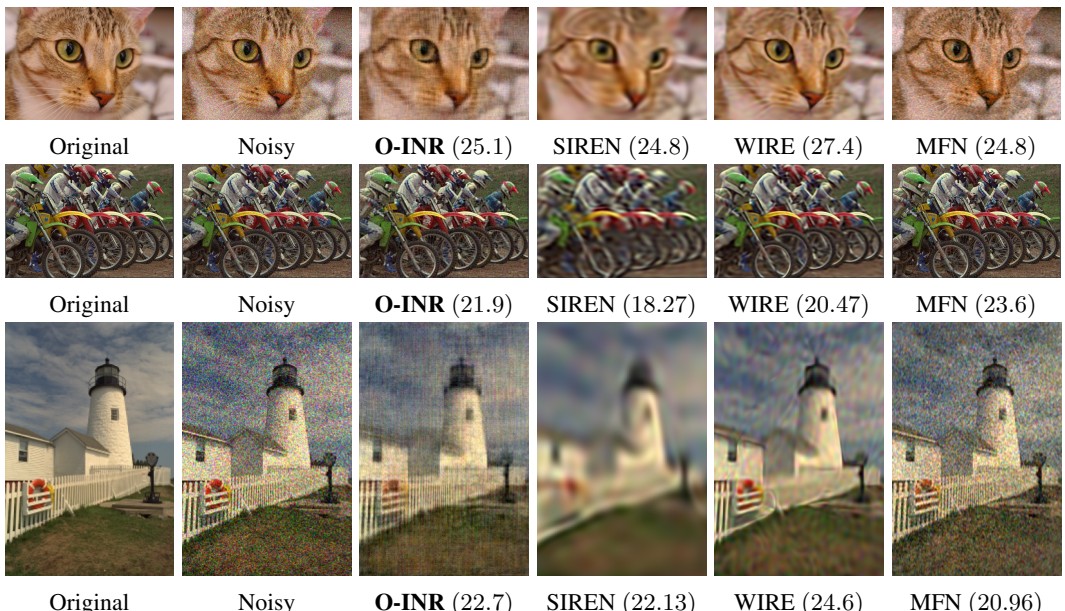

Figure 13: Performance comparisons of O-INR for representing noisy images. For each method, we note the PSNR it achieves on the image in dB. O-INR and other baselines perform similarly.

# F  EFFECTIVENESS OF CONTINUOUS CONVOLUTION IN O-INR

We present results for 2D image representation and 2D image denoising where continuous convolution was used in O-INR. As mentioned in the main paper, continuous convolutions are not strictly necessary for these tasks. Here we demonstrate that performance of O-INR isn't effected by this choice as PSNR is comparable whether one uses continuous or discrete convolution. For example, in Fig. 14, the coffee-mug image attains a PSNR of 31.18 with continuous convolution O-INR whereas with discrete convolution O-INR, PSNR on the same image is 32.04 as reported in Fig. 3.

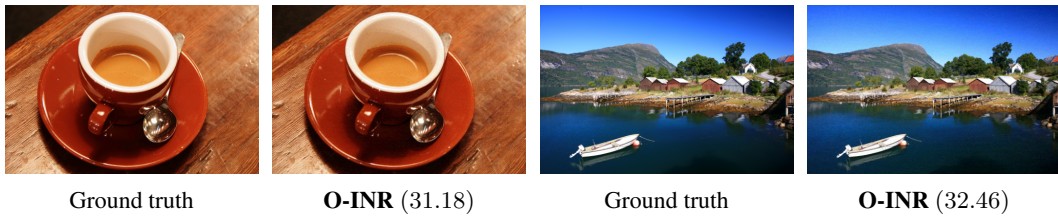

Figure 14: Performance of O-INR for 2D image representation using continuous convolution. The ground-truth together with images from O-INR with the PSNR value in dB. Performance is comparable to O-INR using discrete convolution

Similarly, in the case of representing noisy images, as can be seen in Fig. 15, for the image of an astronaut, continuous convolution based O-INR achieves PSNR of 23.5, on the other hand as presented in the main paper (Fig. 4), with discrete convolution, we can achieve a PSNR of 24.48

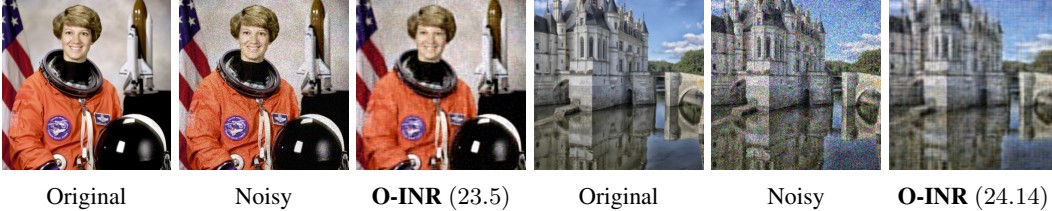

| Original | Noisy | **O-INR** (23.5) | Original | Noisy | **O-INR** (24.14) |

Figure 15: Performance of O-INR with continuous convolution for representing noisy images. We note the PSNR it achieves on the image in dB. Performance is comparable to O-INR using discrete convolution

on the same image. Hence, it is evident that the performance of O-INR is fairly independent of the choice of continuous or discrete convolution and one can choose based on the representation task at hand.

## G    NOISE AS POSITIONAL ENCODING FOR O-INR

As mentioned in **Remark 2** of the main paper, O-INR is capable of simply using noise as a proxy for the high frequency positional encoding term due to the use of convolutional layers. But this is a poor choice for standard INRs with MLP layers due to the lack of location bias. Here we present empirical evidence for the effectiveness of using noise sampled from standard normal distribution as providing high frequency component for the positional encoding as can be seen in Fig. 16

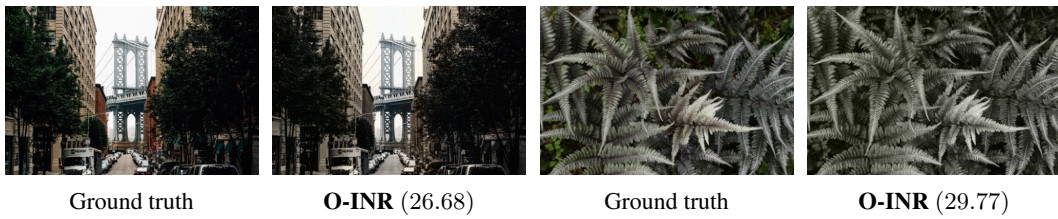

| Ground truth | **O-INR** (26.68) | Ground truth | **O-INR** (29.77) |

Figure 16: Performance of O-INR for 2D image representation using standard normal noise for high frequency positional encoding. The ground-truth together with images from O-INR with the PSNR value in dB.

## H    O-INR FOR SEQUENCE DATA

We used a learning rate of $0.001$ along with 20 positional encodings for each spatial dimension, 10 for $\sin$ and 10 for $\cos$ for training an O-INR model. When trained on the first 16 frames of a cat video Sitzmann et al. (2020), O-INR can achieve an average PSNR of $35.68$ (or MSE of $0.00027$. Additionally we also train O-INR on frames obtained from sub-sampling a video, demonstrating the capability of our method to recover the original sequence despite seeing only a sparse version. Please refer to videos present in the "Result" folder for original and videos recovered from trained O-INR, corresponding to both experimental settings: "consecutive" and "sparse". The "consecutive" sub-folder contains results for the scenario where O-INR was trained on consecutive frames. The folder names therein indicate the video, either of a cat or a road scene and the number denotes the value of 'n' for the first 'n' consecutive frames in the video. The "sparse" sub-folder has results for the scenario, where O-INR was trained on a sparse subset of sub-sampled frames from the video. Folder names therein indicate the dataset, the number of frames used to train the model and the final number of frames present in the video recovered via O-INR.

## I    APPLICATIONS TO BRAIN IMAGING

For the 3D Brain image data, we chose approximately 140 subjects each from the cognitively normal (CN) and diseased (AD) groups and trained O-INR on the T1 MRI scans as mentioned in §6. For performing the group difference analysis using O-INR we use only 34% of available slices, by

sampling every third slice along the Coronal direction for training the O-INRs. We used 20 positional encodings for each spatial dimension - 10 for $\sin$ and 10 for $\cos$. An initial learning rate of $0.001$ was used alongside a Cosine Annealing scheduler with a minimum learning rate of $5e-4$ and maximum steps of $10000$. The whole brain image was generated at the original resolution using the trained model. The models trained achieved an MSE of $2.67e-4$ at the original resolution, indicating that O-INR is able to represent the 3D volume well.

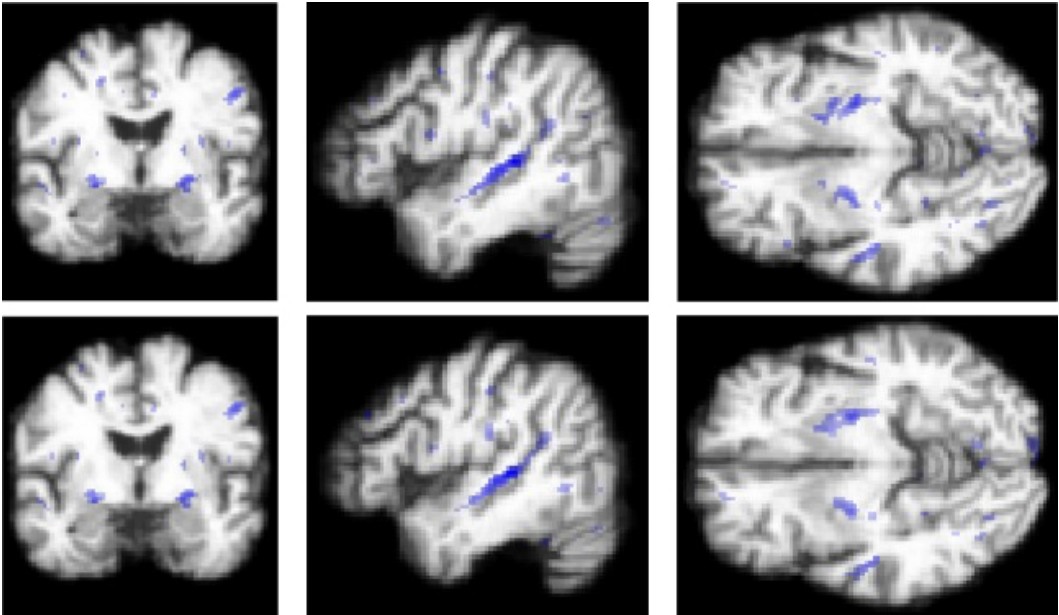

Figure 17: Top and Bottom rows: Overlay of filtered statistic image from group difference analysis of original (full resolution) and images generated via O-INR trained on sparsely sampled (30% slices) AD and CN images respectively. The above images indicate that O-INR is successful in preserving the group difference in 3D brain imaging data.

In order to perform statistical analysis, we used the Statistical non-Parametric Mapping (SnPM) toolbox. We performed statistical group difference (voxel-wise t-test) on the real data (CN versus AD) with 10000 permutations. Then, the same analysis process was repeated on O-INR derived data (CN versus AD). Note that the O-INR's were trained only a fraction of the original resolution. We find that voxels reported to be significant (uncorrected p-values) on the real data analysis agree fully with the analysis results on data based on O-INR slice imputation. Sizable clusters agree although the spatial extent is reduced (higher Type 2 error). This is evident in both the over-lay diagram in Fig. 17 as well as the $T$-statistic and uncorrected $p$-values in Fig. 18.

## J  WEIGHT SPACE INTERPOLATIONS OF O-INR

In addition to performing interpolation between two different O-INRs, we also manipulated convolutional layers in individual O-INRs. We find that manipulating individual convolutional layers by interpolating its weights while holding others fixed yields structurally coherent changes to the image as shown in Fig. 19. In particular, early layers in the O-INR capture large-scale features of the image (Conv1 perturbs the shape of the head, Conv3 perturbs the eyes and nose) while later layers reflect local properties such as color and texture (Conv4 and Conv5). In Fig. 20 we present additional results for weight interpolation between O-INRs trained on images from CelebA Liu et al. (2015).

| T | $p_{uncorr}$ | x,y,z mm | | | T | $p_{uncorr}$ | x,y,z mm | | |
|---|---|---|---|---|---|---|---|---|---|
| 6.00 | 0.0001 | 21 | 48 | 37 | 6.00 | 0.0001 | 21 | 48 | 37 |
| 2.97 | 0.0017 | 22 | 42 | 44 | 3.29 | 0.0002 | 22 | 41 | 44 |
| 1.53 | 0.0650 | 20 | 61 | 28 | 2.76 | 0.0027 | 20 | 57 | 31 |
| 4.23 | 0.0001 | 57 | 59 | 32 | 4.06 | 0.0002 | 72 | 48 | 36 |
| 3.92 | 0.0001 | 55 | 69 | 35 | 1.63 | 0.0542 | 71 | 55 | 32 |
| 3.02 | 0.0016 | 60 | 68 | 42 | 3.97 | 0.0001 | 55 | 69 | 35 |
| 4.06 | 0.0002 | 72 | 48 | 36 | 3.64 | 0.0002 | 57 | 59 | 32 |
| 3.92 | 0.0002 | 65 | 45 | 19 | 3.09 | 0.0012 | 59 | 74 | 41 |
| 2.62 | 0.0055 | 55 | 45 | 15 | 3.92 | 0.0001 | 65 | 45 | 19 |
| 1.96 | 0.0250 | 63 | 37 | 17 | 2.78 | 0.0030 | 65 | 37 | 17 |
| 3.88 | 0.0002 | 61 | 52 | 42 | 2.55 | 0.0057 | 55 | 45 | 15 |
| 3.82 | 0.0002 | 33 | 66 | 40 | 3.88 | 0.0001 | 33 | 66 | 40 |
| 2.78 | 0.0033 | 35 | 71 | 34 | 1.80 | 0.0388 | 37 | 67 | 33 |
| 3.68 | 0.0002 | 33 | 61 | 33 | 3.69 | 0.0002 | 29 | 92 | 49 |
| 3.48 | 0.0004 | 67 | 70 | 58 | 2.89 | 0.0026 | 33 | 85 | 45 |
| 2.11 | 0.0185 | 60 | 72 | 54 | 2.24 | 0.0126 | 35 | 97 | 46 |
| 3.48 | 0.0006 | 40 | 20 | 24 | 3.42 | 0.0003 | 61 | 51 | 43 |
| 3.38 | 0.0006 | 71 | 60 | 56 | 1.87 | 0.0315 | 62 | 58 | 37 |

Figure 18: Statistical analysis results from SnPM showing $T$-statistics, uncorrected $p$-values and the cluster center location of sizable clusters. Table on the left summarizes the analysis results on the real data whereas the table on the right summarizes the analysis on the data from O-INR slice imputation. Nearly all sizable clusters (in bold) on the left have a corresponding cluster on the right (the rank may be slightly up or down) indicating strong agreement between the results.

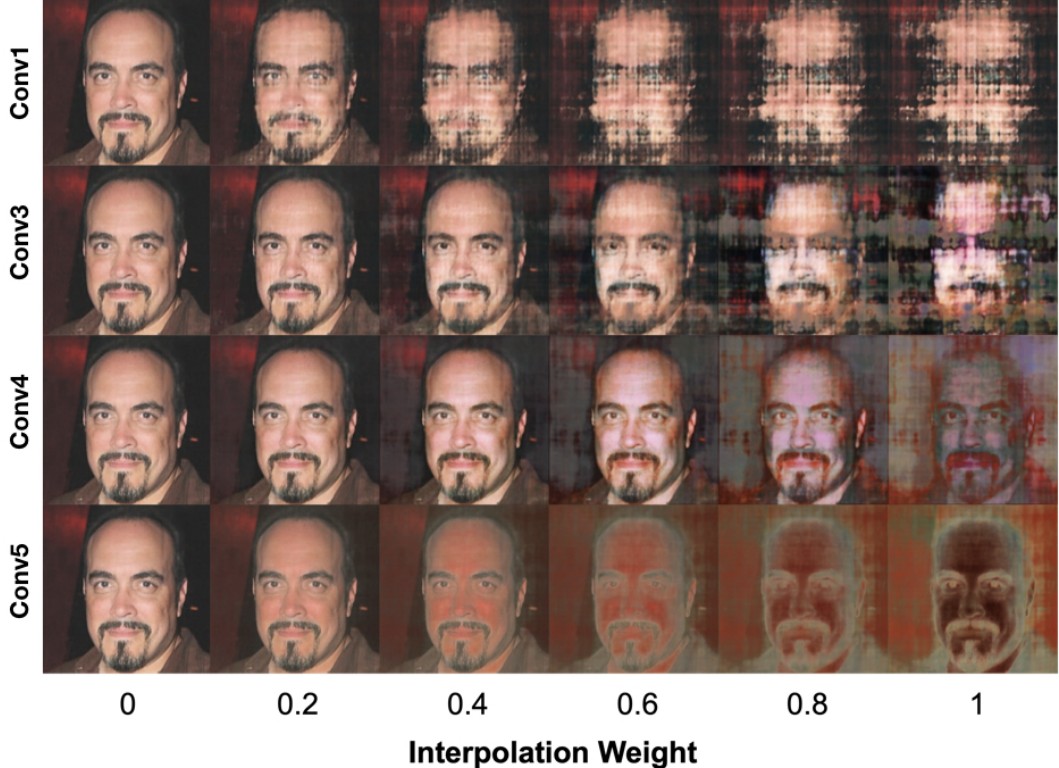

Figure 19: Images produced by interpolating the weights of a single convolutional layer between two O-INRs fit on different CelebA images. Other weights are held fixed.

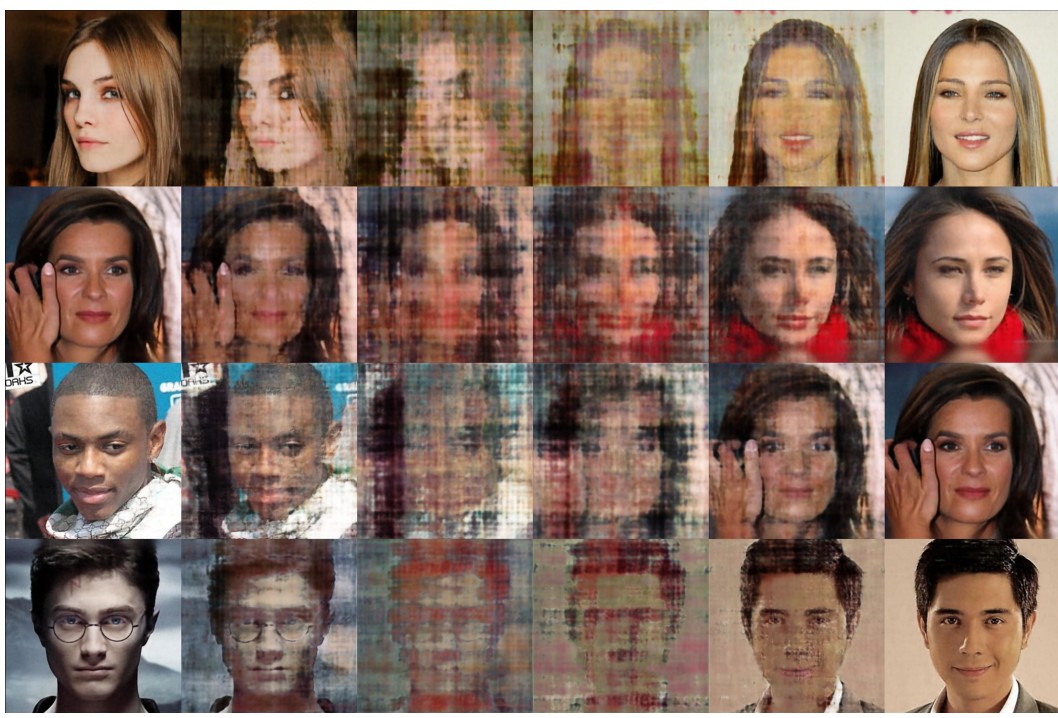

Figure 20: Randomly selected examples of images produced by interpolating the weights of O-INRs fit on different CelebA images.

