# OpenReview forum: "Operator-theoretic Implicit Neural Representation"
_ICLR.cc/2024/Conference — ICLR 2024 Conference Withdrawn Submission_

### Official Review · Reviewer_255q · 2023-10-26

**Soundness:** 2 fair
**Presentation:** 2 fair
**Contribution:** 2 fair
**Rating:** 3
**Confidence:** 3

**Summary:**

The main contribution of this paper is a new INR architecture, called Operational INR (O-INR). This architecture is introduced as a mapping from one function space (the positional encoding function space) to another (the INR function space). The mapping is performed using integral transforms. The proposed O-INR $h(\omega)$ for the positional encoding $f(\omega)$ is $h(\omega)=\int_{\omega'\in\Omega}K_\phi(\omega,\omega')f(\omega')d\omega'$, where the integral kernel $K_\phi(\omega,\omega')$ is proposed to be a convolutional kernel parameterized by $\phi$. Two key differences with typical INRs are 1) the use of convolution rather than MLP and b) O-INR computes a transform over the full domain rather than in a single point. Other methodological contributions of the paper include the inclusion of of high-frequency noise in the positional embedding, the use of O-INR to compute derivative of the encoded function, a dedicated formulation to handle the temporal domain. O-INRs are evaluated in multiple experiments: image representation and denoising, multiresolution training on images, occupancy map encoding, video encoding, brain imaging (encoding ang missing slice extrapolation). Derivative computation and weight interpolation are also evaluated.

**Strengths:**

originality:
- The proposed approach is very original. It defers significantly from the different approaches to implicit neural representation. The experiments also highlights these differences. The brain imaging experiment is also quite original.

quality:
- There are a lot of different experiments.

clarity:
- The paper is mostly clear when describing the approach.

significance:
- I believe this paper proposes interesting ideas that seem promising.

**Weaknesses:**

quality:
- In experiment 4.1, the text claims that "O-INR" is comparable/better than baselines but it achieves the worse PSNR (figure 2).
- I found the baselines a bit lacking. Only 3 baselines are used: SIREN (2020), WIRE (2023) and MFN (2021). There is no comparison to grid-based INR such as instant-NGP in any experiment. These 3 baselines are only present in some experiments, the image ones, occupancy map encoding and weight interpolation. Unless I missed it, no baseline is considered in the video experiment and the brain imaging one, arguably the most complex ones.
- The video experiment is performed on a single video. Typically, experiments are conducted on datasets such as UVG. This might be a computational limitation, but then it would make sense to use a video from a commonly used dataset.
- Tables 2 and 3 are badly positioned, with a single line of text below or above them.
- Table 1 is very close to the text.

clarity:
- The paper describes the proposed approach as a mapping between two function spaces and suggest the mapping is done between different $f$ and yields different $h$, see for example figure 1, the two paragraphs above equation 2 and the paragraph below. However, it is common for INR networks to use the same mapping to represent every signal and as far as I could tell the same positional encoding is also used for each signal of the same class in the experiments of the paper. So I found this discussion over multiple positional encoding functions a bit confusing.
- Reinforcing this confusion is equation (3), where the domains of the transformation function are denoted as a single function ($f \rightarrow h$) whereas function spaces are defined above.
- It is not clear for me what is the "location bias" mentioned after equation 6.
- The temporal representation is discussed in the experimental setup of one experiment (equations 7 and 8). I also could not understand it completely: the embedding functions are modified (the same power of 2 is used in all terms) and an offset parameterized by 2 parameters and depending on the power of 2 is introduced. To me, these choices are neither motivated nor explained in the paper.

significance:
- There is no ablation study to show the difference between convolutional and MLP kernel. Hence it is no clear to me whether both using a convolutional kernel or computing a transform over the whole domain signal brings benefits. This point is also not discussed as far as I could tell.
- Same comment for the high-frequency noise and the special temporal modeling.

Furthermore, I would like to bring up the following details:
- Appendices number are often not mentioned in the text when referring the reader to the appendix, making it difficult to read the paper.

**Questions:**

I would welcome the authors' opinion about the weaknesses listed above. In addition, I have the following question:
- The paper highlights the ability of O-INR to compute derivatives. I know a few references are provided, but could you please explain in a few words the importance/advantages of this ability?

---

### Official Review · Reviewer_x28W · 2023-10-27

**Soundness:** 2 fair
**Presentation:** 1 poor
**Contribution:** 2 fair
**Rating:** 1
**Confidence:** 4

**Summary:**

The paper proposes an implicit neural representation that transforms the positional encoding functions of coordinates into the signal function. Instead of using an MLP to evaluate the INR point by point, continuous convolution is used to take into account the entire positional encoding functions without compromising the capability of free query in the domain.

**Strengths:**

- Originality: The paper provides an alternative viewpoint for implementing INRs with intuitions from neural operators to deal with continuous output function.
- Significance: The paper

**Weaknesses:**

- Lack of important elements about the model: One critical aspect missing from the model description is a clear presentation of the network architecture. Equations (4)-(6) seem to suggest a linear transformation between the positional encoding function $f$ and the target $h$, which implies that the neural network consists of just one linear layer without any nonlinear activation functions. However, in the comparison with other INRs, the authors highlight the advantages of kernels over an MLP. To eliminate any ambiguity, it's essential to provide more detailed information about the actual network architecture.
- Lack of details on the implementation of baseline methods: It is challenging to assess the performance gain without a comprehensive description of the implemented baseline methods, including details such as the number of parameters, the number of layers, and the coordinate range. Additionally, there are two implementations for MFN; it would be helpful to specify which one was chosen for this evaluation.
- Given the minor result differences between the 2 best methods, assessment solely based on individual images no longer offers informative comparisons with baseline models. To establish the method's credibility, it is recommended to report statistical performance metrics on a reasonably sized dataset. Based on the current results, it's challenging to determine if the model performs as claimed.
- In Section 5, I couldn't locate any comparisons with the baseline methods. If it's necessary to reference other papers to make these comparisons, it indicates that the paper is not self-contained and could benefit from better organization and inclusion of baseline comparisons within the document.
  Also, the authors critique the use of mapping with $t$ as coordinates in other MLP-based INRs. However, without ablation studies, it's challenging to understand the motivation behind this criticism, even if it is reasonable.
- In Section 7, I find it difficult to accept the claim that gradient evaluation cannot be accomplished with baseline INRs. With the availability of autodiff libraries (e.g. JAX, or even PyTorch), it is possible to perform gradient evaluation, which should be acknowledged and discussed in more detail.
- I believe that the motivation for the interpolation task in Section 8 should be clarified. Furthermore, since it's observed that linearly interpolating the weight results in a linear interpolation of the image, it would be helpful to explain the significance and implications of these findings.

**Questions:**

- I believe the paper could significantly benefit from a more thorough clarification of the details of the model and baselines, and a more convincing comparison with the baselines.
- Instead of prioritizing numerous applications with limited comparisons, it would be more effective to emphasize how the operator viewpoint can enhance the representation of a set of signals, showcasing the method's potential benefits and significance.

---

### Official Review · Reviewer_LTrk · 2023-10-30

**Soundness:** 2 fair
**Presentation:** 1 poor
**Contribution:** 1 poor
**Rating:** 3
**Confidence:** 3

**Summary:**

This paper proposes a framework where instead of using INR, they use (continuous) convolution layers applied to the positional encoding functions. They provide numerical experiments, highlighting its performance on INR benchmark tasks, i.e.  regression to images, denoising, representing 3d volumes etc..

**Strengths:**

The experimental analysis seems quite extensive, and a broad range of tasks where tackled.

Interesting to see someone questioning the very essence of INRs.

**Weaknesses:**

* Section 3 and presentation of the contribution is very much unclear and poorly written. The function space is not clearly defined. What is the space of sinusoidal positional encodings? This should clearly be defined. You should also mention that the space you consider in practice is a discrete, finite space.

* Maybe I am missing something here but I feel that the contribution is rather limited. I feel that the solution is not well motivated, and the experiments aren't entirely convincing. The related work should be a place to *clearly* state the difference of your method wrt to previous methods, ie continuous convolutions, or for instance missing work on neural operators. I do not think that bringing continuous convolutions in the context of INRs is very much of a contribution in itself.

* no code

**Questions:**

In practice, the signal has to be discretized. How is it discretized? What grid do you use? What errors are we making?

Do you ever use integral operators that are not convolutions? If not, why present integral operators?

What is the motivation behind applying convolutions on sinusoidal positional encodings?

If your related work is about continuous convolutions, i feel that it is only normal to compare against a continuous convolutional network (that takes coordinates as input for example, or even a constant).

---

### Official Review · Reviewer_mmzV · 2023-11-02

**Soundness:** 2 fair
**Presentation:** 3 good
**Contribution:** 2 fair
**Rating:** 5
**Confidence:** 4

**Summary:**

This paper presents an operator-theoretical reformation of Implicit Neural Representation (INR). Instead of viewing INR as a pointwise mapping from position embedding to coordinate-aligned value, the authors regard INR as an operator that maps a predefined function space to the signal space via integral transforms. To improve efficiency and numerical stability, the authors propose to use convolution as a surrogate of the general integral transforms.

**Strengths:**

+ The paper is well-written, nicely structured, and easily comprehensible. The problem setting is highly relevant, as it endeavors to represent a set of signals through INR, enhancing the practical utility of this approach.

+ The proposed method introduces a novel reformulation that, to the best of my knowledge, is the first to regard INR as a transformation between function spaces. This reformulation, in turn, offers insights into numerous convolution-based architectures that employ position embeddings as inputs, as seen in [1].

+ The experimental section encompasses eight proof-of-concept experiments for INRs. These experiments span a wide spectrum of potential INR applications, underscoring the versatility and adaptability of this approach.

[1] Karras et al., Alias-Free Generative Adversarial Networks

**Weaknesses:**

- While the operator perspective is indeed novel, it seems that the method essentially involves applying convolution on a position-embedding lattice, if I understand correctly. This computational paradigm is already established in various works, as evidenced by references such as [1] and [2].

- Arguably, one of the most significant advantages of INR is its ability to decode arbitrary points without the need for neighborhood sampling. This property is particularly valuable for unstructured decoding, a common requirement in 3D domains, as seen in surface regression through point clouds [3] or ray-based rendering in NeRF [4]. However, as I understand it, there may be doubts regarding the efficiency of O-INR in achieving this, as it appears that O-INR requires local window sampling to perform convolution for pointwise value computation.

- The experiments presented, while covering a variety of applications, may not fully support the main claim of "from one signal to a set of signals." This claim seems to be substantiated merely by Sec. 5 and 6, without direct comparisons to relevant baselines, such as Functa [5]. Furthermore, in Sec. 7, it might be inadequate to solely focus on derivative computation, as there is a need to consider a broader range of tasks, as introduced in [6] and [7].

[1] Karras et al., Alias-Free Generative Adversarial Networks

[2] Wang et al., Patch Diffusion: Faster and More Data-Efficient Training of Diffusion Models

[3] Sitzmann et al., Implicit Neural Representations with Periodic Activation Functions

[4] Mildenhall et al., NeRF: Representing Scenes as Neural Radiance Fields for View Synthesis

[5] Dupont et al., From data to functa: Your data point is a function and you can treat it like one

[6] Xu et al., Signal Processing for Implicit Neural Representations

[7] Navon et al., Equivariant architectures for learning in deep weight spaces

**Questions:**

1. The implementation of continuous convolution, as mentioned in Sec. 3 and 4, remains unclear. Is it exactly the same with the approach described in [1]?

2. Regarding the efficiency of convolution parameterization, it is essential to clarify why this approach is more efficient compared to alternatives. For instance, if pointwise mapping, as seen in SIREN [2], can be considered a form of 1x1 convolution, should it not be more efficient?

3. The assertion that derivative computation is “a functionality not currently available otherwise” requires further clarification. To my knowledge, derivatives can be computed in closed form through standard INRs, as demonstrated in [2], [3], and [4]. Additionally, the claim that "O-INR allows seamless calculus operations" prompts the question of whether O-INR can efficiently compute integrals as well.

[1] Romero et al., Continuous Kernel Convolution For Sequential Data

[2] Sitzmann et al., Implicit Neural Representations with Periodic Activation Functions

[3] Lindell et al., AutoInt: Automatic Integration for Fast Neural Volume Rendering

[4] Xu et al., Signal Processing for Implicit Neural Representations